# Genome-Wide Association Study of the Reproductive Traits of the Dazu Black Goat (*Capra hircus*) Using Whole-Genome Resequencing

**DOI:** 10.3390/genes14101960

**Published:** 2023-10-19

**Authors:** Xingqiang Fang, Bowen Gu, Meixi Chen, Ruifan Sun, Jipan Zhang, Le Zhao, Yongju Zhao

**Affiliations:** 1College of Animal Science and Technology, Southwest University, Chongqing 400715, China; 15881044561@163.com (X.F.); gubw04@163.com (B.G.); chenmeixi529@163.com (M.C.); sunruifan111@163.com (R.S.); jpanzhang@live.com (J.Z.); zhaole0228@163.com (L.Z.); 2Chongqing Key Laboratory of Herbivore Science, Chongqing 400715, China; 3Chongqing Key Laboratory of Forage & Herbivore, Chongqing 400715, China

**Keywords:** goat, reproductive traits, udder traits, whole-genome resequencing, GWAS

## Abstract

Reproductive traits are the basic economic traits of goats and important indicators in goat breeding. In this study, Dazu black goats (DBGs; n = 150), an important Chinese local goat breed with excellent reproductive performance, were used to screen for important variation loci and genes of reproductive traits. Through genome-wide association studies (GWAS), 18 SNPs were found to be associated with kidding traits (average litter size, average litter size in the first three parity, and average litter size in the first six parity), and 10 SNPs were associated with udder traits (udder depth, teat diameter, teat length, and supernumerary teat). After gene annotation of the associated SNPs and in combination with relevant references, the candidate genes, namely *ATP1A1*, *LRRC4C*, *SPCS2*, *XRRA1*, *CELF4*, *NTM*, *TMEM45B*, *ATE1*, and *FGFR2*, were associated with udder traits, while the *ENSCHIG00000017110*, *SLC9A8*, *GLRB*, *GRIA2*, *GASK1B*, and *ENSCHIG00000026285* genes were associated with litter size. These SNPs and candidate genes can provide useful biological information for improvement of the reproductive traits of goats.

## 1. Introduction

Goats are one of the most important livestock in China, and the goat industry plays an important role in improving the national economy. According to the report of the Ministry of Agriculture, there are nearly 120,000 Dazu black goats (DBGs) in southwest China. It is an important local goat breed in Chongqing and has been widely farmed because of its excellent reproductive performance. The birth rates of twins, triplets, and quadruplets can reach 47.27, 40.02, and 10.29%, respectively, in this breed [1], and according to the National Animal and Poultry Genetic Resources Variety list (2021 Edition), the kidding rate of primiparous does of Dazu black goat is 218%, and the kidding rate of multiparous does is 272%. Thus, it has high development value and application potential.

Reproductive characteristics are important economic traits in the goat industry, and improving the kidding traits of goats is particularly important for increasing the economic benefits of the goat industry. Litter size is a very important economic trait, and is affected by many factors, including ovarian follicular development, ovulation, fertilization rate, embryonic development, and embryo implantation [2], and litter size in goats is a low heritability trait and varies from one goat breed to another. Odubote’s [3] study found the heritability for litter size in West African dwarf goats to be 0.35 ± 0.05 and 0.32 ± 0.07. In addition, Bangar’s study showed that the heritability of litter size in Beetal goats was 0.08 [4]. Although the *FecB* gene has been identified as a high-breeding gene for sheep, goats still do not have a recognized candidate gene with a strong correlation with kidding traits. So, to understand the genetic basis of litter size traits, many molecular biology and genetic studies have been carried out to detect chromosome regions, candidate genes, and genetic markers associated with reproductive performance in goats [5,6,7]. Genome-wide association study (GWAS) is a method of correlating millions of single-nucleotide polymorphisms (SNPs) in the genome as molecular genetic markers at the genome-wide level, followed by comparisons to discover genetic variants that affect complex traits [8]. Previously, GWASs have been widely used in studies of goat litter size [6,9], horn status [10], body conformation [11], and milk quality [12].

Many candidate genes and genetic markers are associated with reproductive traits in mammals. Islam et al. [13] found several candidate genes related to goat fecundity in a Chinese goat population. These include the *MARF1*, *SYCP2*, *TMEM200C*, *SF1*, *ADCY1*, and *BMP5* genes. Xu et al. [9] conducted multiple independent GWASs in five sheep breeds of high prolificacy and one of low prolificacy, and identified that the *BMPR1B*, *FBN1*, *MMP2*, *GRIA2*, *SMAD1*, and *CTNNB1* genes were associated with litter size. To study the genomic regions related to the teat and udder structure of Canadian Angus dairy cows, Devani et al. [14] conducted a weighted single-step GWAS on the teat and udder scores of 1582 cows, and identified a total of 94 and 71 genes in the regions related to teat and udder scores, respectively. In addition, Abdoli et al. [15] conducted a genome-wide association study (GWAS) on the estimated breeding value (EBV) of first birth age and lambing interval of Lori–Bakhtiari ewes. Two SNPs associated with lambing interval were found on chromosomes 1 and 2.

To better understand the genomic regions and biological pathways affecting the reproductive traits of DBGs, we performed a genome-wide association analysis of seven reproductive traits. The SNPs and candidate genes identified in this study can provide useful biological information for improvement of the reproductive traits of goats.

## 2. Materials and Methods

### 2.1. Ethics Statement

The Chongqing Key Laboratory of Forage and Herbivore approved this experiment. All goat experiments followed the Institutional Animal Care and Use Committee of Southwest University regulations (IACUC-20230306-16), and no animals were anesthetized or euthanized during this study.

### 2.2. Phenotypic and Genotypic Data

In this study, a total of 150 female DBGs aged 3–5 years and weighing 35–40 kg were selected from Tengda Animal Husbandry Co., Ltd. (Chongqing, China). We also collected data on litter size (LS) and used Microsoft Excel to calculate the average litter size (ALS), average litter size in the first three parity (ALS3), and average litter size in the first six parity (ALS6), and the udder traits of 80 does in constant lactation period were measured:(1)Udder depth (UD) is the vertical distance from the root of the left udder to the top of the teat.(2)Teat diameter (TD) is the teat diameter of the left udder.(3)Teat length (TL) is the teat length of the left udder.(4)Supernumerary teats (SNTs) were observed and recorded with the naked eye.

The genome data used in this study were generated as part of a previous whole-genome resequencing study, where the sequencing of each sample was performed on the DNBSEQ-T7 platform (Complete Genomics and MGI Tech, Shenzhen, China) [16]. The average sequencing depth of each sample was about 6×, and the average mapping rate was 99.06%.

### 2.3. Statistical Analysis

Standard quality control of genotypes and phenotypes is needed in GWAS to eliminate the influence of abnormal genotypes and phenotypes on the results of the analysis. The genotype QC process removes individuals or loci that meet the following criteria: (1) deletion rate > 10%; (2) minor allele frequency (MAF) < 1%; and (3) Hardy–Weinberg equilibrium (HWE) with *p* < 1 × 10^−6^. Phenotype quality control criteria: delete individuals with phenotypic deletion [16]. Since GWAS does not allow for missing or unknown SNPs, we used Beagle 5.0 [17] to infer missing data. The GWAS model used in this study was a mixed linear model. The model was first published in Nature Genetics by Yu et al. [18] and is widely used because it can correct for population structure and the complex genetic relationship within a population. The model is as follows:y = Xβ + Z_kγk_ + ξ + e
where y is the phenotype; Xβ is the population structure effect and fixed effects, such as year and season; Z_kγk_ is the marker effect to be tested; ξ~N (0, Kϕ^2^) is the polygenic effect; e~N (0, *Iσ*^2^) is the residual effect; and K is the kinship matrix inferred from the SNPs.

Then, we used GEEMA to obtain significant SNPs, and the *p*-value corrected using Bonferroni was 6.01 × 10^−8^. Significant SNPs were visualized as a Manhattan map and quantile–quantile (QQ) map in R (v4.0.4) [19].

Candidate genes were identified according to their physical positions and functions based on the ARS1 reference genome assembly, and the significant SNPs were annotated to the corresponding genes using ANNOVAR (http://www.openbioinformatics.org/annovar/, accessed on 14 September 2023) [20]. GO enrichment is an international standardized classification system of gene function, with a total of three ontologies describing the molecular function (MF), cellular location (CC), and participating biological process (BP) of genes. The KEGG database is the main public database of genetic pathways. The pathway significance enrichment analysis takes the pathways in KEGG as a unit and uses a hypergeometric test to identify the pathways that are significantly enriched in candidate genes compared with the whole reference genome. Both GO enrichment and KEGG pathway analysis of candidate genes were performed using clusterProfiler in R (v4.0.4).

## 3. Results

### 3.1. Phenotypic Data Analysis

The descriptive statistics for the seven reproductive traits (ALS, ALS3, ALS6, UD, TD, TL, and SNT) measured in this study are presented in Table 1. To ensure the reliability of the results for follow-up analyses, we estimated the standard deviations, confidence intervals, and coefficients of variation. The dependent variables (ALS, ALS3, etc.) were approximately normally distributed (Appendix A Appendix A). In addition, a total of 20 individuals in the selected group had supernumerary teat, with a frequency of 25%.

### 3.2. Genome-Wide Association Study

The QQ plot in Figure 1 shows the observed and expected *p*-value distribution of reproduction traits found in the GWAS. Most of the SNPs did not deviate from the expected *p*-value, and only a few SNPs in ALS6 and TL showed a certain degree of deviation. After screening for and obtaining the adjusted phenotype, GWAS was performed on the identified SNPs. Figure 2 and Figure 3 and Table 2 show the Manhattan diagrams of the evaluated reproductive traits. ALS identified a locus on chromosome 5 and annotated two genes, *TBC1D22A* and *TAFA5*, near the locus. ALS6 identified 16 important SNPs, and among them, eight significant differences were identified on chromosome 17, most of which were annotated to the *GLRB* gene. Additionally, one locus was identified on each of the chromosomes 6, 11, 13, 19, 27, and 28. UD identified a locus on chromosome 3 and annotated two genes *ATP1A* and *ENSCHIG00000022779* near the locus. TD identified one locus on chromosomes 15 and 18, and annotated the *LRRC4C* and *ENSCHIG00000000547* genes near it, respectively. TL identified one locus on chromosome 24 and 29 respectively, and two loci on chromosome 15. The loci on chromosome 15 were annotated to the *XRRA1*, *NTM*, and *TMEM45B* genes, while the locus on chromosome 24 was annotated to the *ENSCHIG00000022267* and *CELF4* genes. The locus on chromosome 29 was annotated to the *SPCS2* gene. SNT identified one locus each on chromosomes 18, 23, and 26. The locus of chromosome 18 was annotated to the *CD83* and *ENSCHIG00000008871* genes, while the locus of chromosome 23 was annotated to the *ATE1* and *FGFR2* genes. The locus of chromosome 26 was annotated to the *BICRA* gene.

### 3.3. GO Enrichment and KEGG Analysis

A total of 217 different GO entries were enriched in the selected reproduction-related candidate genes. ALS6 alone was enriched to 77 strips, while SNT, TD, UD, and TL were enriched to 63, 52, 17, and 8 strips, respectively. A large number of items were enriched in molecular functions and biological processes (Appendix A). The top 10 ALS6 enrichment items were as follows: transmitter-gated ion channel activity, transmitter-gated channel activity, neurotransmitter receptor activity, extracellular ligand-gated ion channel activity, ligand-gated ion channel activity, ligand-gated channel activity, inorganic molecular entity transmembrane transporter activity, ion transmembrane transporter activity, gated channel activity, and ion channel activity (Figure 4).

KEGG analysis revealed a total of 40 significantly enriched pathways (Appendix A). UD enriches the most significant enrichment pathways, among which proximal tubule bicarbonate reclamation, aldosterone-regulated sodium reabsorption, and carbohydrate digestion and absorption were the most significant. Importantly, all of the UD-enriched pathways are related to the physiological function of the udder. In addition, the SNT enrichment pathway was mostly related to cancer and stem cells, which is noteworthy.

## 4. Discussion

### 4.1. Kidding Traits

As litter size is an important limiting factor affecting the reproductive efficiency of goats, the identification and use of goat molecular markers to increase litter size can effectively accelerate the development of the goat breeding industry. In the study of E et al. [21], genome-wide selection signatures analysis of two groups of 31 DBGs with different kidding performances revealed two new candidate genes, *LRP1B* and *GLRB*, related to litter size. In addition, they also analyzed the effect of copy number variation (CNV) on the litter size of DBGs and found that the new CNV of the *CBLB* gene was significantly related to the litter size of DBGs [5]. Wang et al. [7] divided 40 Jining gray goats into three groups with different kidding performances according to the litter size in the first parity. Whole-genome sequences were analyzed using Fst, π, and Tajima’s D, and a total of 111 genome-wide selection regions and 42 genes related to litter size were identified. Based on these findings, this study selected the average litter size of multiple parities (ALS, ALS3, and ALS6) for analysis, hoping to improve the accuracy of previous studies.

According to the National Animal and Poultry Genetic Resources Variety list (2021 Edition), there are eight goat breeds with an average kidding rate of more than 200%, one of which is DBG. The reproductive period of female DBGs is mostly between 5 and 7 years, while some high-yielding does can even reach eight years. The results of this study showed that a total of 1, 0, and 17 SNPs were associated with ALS, ALS3, and ALS6, respectively. These SNPs are mainly concentrated on chromosome 17, of which six SNPs are located within the *GLRB* gene. This gene is consistent with the candidate gene obtained by E et al. [21] using comparative genomics for small groups of DBGs with different kidding performances. In this study, only one SNP was found to be associated with ALS, and the *TBC1D22A* and *TAFA5* genes annotated by the SNP did not correlate with goat reproductive performance. However, as a member of the TAFA family, *TAFA5* can encode small secretory proteins in the central nervous system and has been shown to increase expression in human malignant tumors [22]. Paulsen et al. [23] and Bundzikova et al. [24] examined the expression pattern of mRNA in rat brains using in situ hybridization and immunohistochemistry and confirmed that *TAFA5* was mainly expressed in the central nervous system, is co-located with vasopressin and oxytocin in neurons, and may be involved in the regulation of homeostasis.

In this study, the significant SNPs associated with ALS6 were annotated to 14 genes. Among them, five genes including *NDST3*, *GLRB*, and *TRAM1L1* have been associated with reproductive traits of livestock. One study investigated the role of heparan sulfate polysaccharide (HSPG) in regulating uterine receptivity by targeting *NDST3* deletion in mice. Results showed that although reproductive tract development and ovarian function appear to be normal in knockout dams, they remained infertile due to implantation defects [25]. Palleria et al. also showed that *NDST3* can promote the synthesis of heparan sulfate (HS) and assist in embryo implantation [26]. In addition, some studies have found that *SLC9A8* is one of the regulatory genes of the sodium/proton reverse transporter (NHE) protein [27]. The NHE protein is a membrane protein found in most organisms, where mice without NHE8 will have no acrosome, abnormal mitochondrial distribution, and decreased motor ability, resulting in infertility [28]. *TRAM1L1* has been associated with fecundity in Holstein cows using GWAS [29], while *GLRB* was shown to be located in the acrosome region of spermatozoa [30]. Previous studies also showed that *GLRB* is related to the litter size of DBG [21]. Finally, *GRIA2* is part of the secretion mechanism of gonadotropin-releasing hormone (GnRH) [31], which affects the secretion of GnRH by regulating calcium ions in GnRH neurons [32,33,34]. There are also different research results: In the study of Sun [35], through GWAS of four goat breeds in southern China, it was found that three SNPs located in *GLRB* and *GRIA2* genes were related to the hind leg hair traits of southern Chinese goats. In another study, Mahmoudi et al. [36] performed a GWAS on Markhoz goats (n = 136) and found that four SNPs on chromosomes 2, 20, and 21 were identified as significantly correlated with litter size. In the present study, the SNPs associated with lambing number were mainly on chromosome 17. It is important to note that in GWAS of litter size, correctly calculating the average parity of does by considering more than three parities is imperative.

### 4.2. Udder Traits

In female mammals, the mammary gland changes dynamically throughout the reproductive cycle, which can be divided into five stages: embryo, prepuberty, puberty, pregnancy, lactation, and degeneration. This developmental process is mainly regulated by hormones, growth factors, and cytokines. Jena et al. [37] recorded the morphological characteristics of the udder (length, column width, row width, depth, circumference, volume) and teat (length, diameter, circumference, height from the ground, inter-teat distance) of goats alongside daily milk yield over three months. Correlation analysis showed that all the teat traits were positively correlated with daily milk yield. Erduran et al. [38] also found that there was a strong correlation between milk yield and breast volume in goats.

Due to the limited number of teats, most does struggle to feed multiple kids. Therefore, teat number is an important consideration in goat breeding, playing a key role in the healthy growth of kids. In addition, supernumerary teats greatly affect the efficiency of machine milking, but Hardwick et al. [39] showed that supernumerary teats may be susceptible to bacterial ingress through the teat canal, and this may have a negative impact on production performance. Ghaffarilaleh et al. [40] found that among the eight different breeds of goats in Iran, Bush, and Boer goats had the highest frequency of SNT. Martin et al. [41] carried out GWAS on the number of accessory nipples in Sanan and Alpine goats and identified 17 regions on chromosome 10 significantly different at the chromosome level, suggesting that SNTs are polygenically inherited.

DBGs not only maintain high reproductive performance, but also ensure a high survival rate of young kids, which is strongly related to the udder function of does, in theory. In this study, the udder traits of 80 lactating does were selected for GWAS. In this GWAS, 1, 2, 4, and 3 important SNPs were detected in UD, TD, TL, and SNT, respectively, and a total of 15 candidate genes (*ATP1A1*, *LRRC4C*, *XRRA1*, *TMRM45B*, *BICRA*, and *CD83*, etc.) were annotated. Liu et al. [42] analyzed the *ATP1A1* of Holstein cows (n = 320) and found that *ATP1A1* was associated with somatic cell score and 305d milk yield, which played a role in mastitis. The candidate genes that may affect SNT are *BICRA*, *CD83*, *ATE1*, and *FGFR2*. In previous studies, Scott et al. [43] found that the lack of haploid in *BICRA* led to a unique neurodevelopmental disorder. *CD83* is a member of the Ig superfamily. Expressed in the form of membrane binding or soluble, it can be detected in a variety of activated immune cells. It is reported that *CD83* participates in the regulation and balance of the immune system and can be used as an immunomodulator with therapeutic potential [44]. In addition, Aalia et al. [45] found that *CD83* promoted the growth and proliferation of ovarian cancer cells. In previous studies, *FGFR2* has mainly been related to breast cancer. Bao et al. [46] showed that *GH* overexpression may promote breast proliferation through *FGF7* and *FGFR2*, and *FGFR2* was expressed in endometrial epithelium and trophoblastic epithelium [47]. Lei et al. detected the mutation of *FGFR2* in the mammary gland of a mouse model and found that FGF/FGFR2 drives the development of breast cancer, and inhibiting *FGFR2* and blocking immune checkpoints can reduce the number of cancer cells [48]. Additionally, Justyna et al. [49] found that *FGFR2* plays a crucial role in normal uterine development and normal uterine function in mice after birth. Arginyltransferase 1 (*ATE1*) is an evolutionarily conserved eukaryotic protein located in the cytoplasm and nucleus. It can regulate metabolism and apoptosis, resulting in decreased intercellular contact and chromosome aberration [50].

The studies mentioned above have made different efforts in exploring the factors affecting reproductive traits in goats, but there are differences in the results of the studies, and these differing findings suggest that reproductive traits in goats are influenced by complex genetic mechanisms. Subsequent studies will explore the function of this candidate gene and the mechanism of its effect on reproductive traits.

## 5. Conclusions

In this study, the genome and reproductive phenotypic data of 150 DBGs were collected and combined with GWAS. Results revealed that the genes *ATP1A1*, *BICRA*, *CD83*, *ATE1*, *NDST3*, *GLRB*, *GRIA2*, and *GASK1B* may be involved in the regulation of reproductive traits in DBGs. The significant SNPs and genes identified in this study will aid in the molecular-based breeding of goats to improve reproductive performance.

## Figures and Tables

**Figure 1 genes-14-01960-f001:**
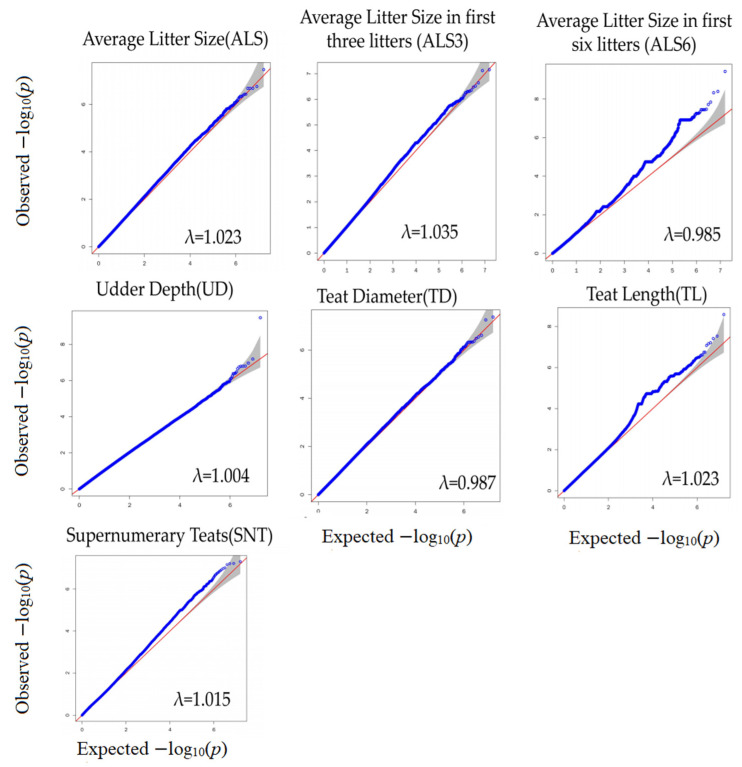
QQ plots of the reproductive traits (ALS, ALS3, ALS6, UD, TD, TL, and SNT) drawn by the expected *p*-value (the uniformly distributed quantile from 0 to 1) and observed *p*-value for each SNP. The shaded parts are the confidence intervals. λ: genomic inflation factor.

**Figure 2 genes-14-01960-f002:**
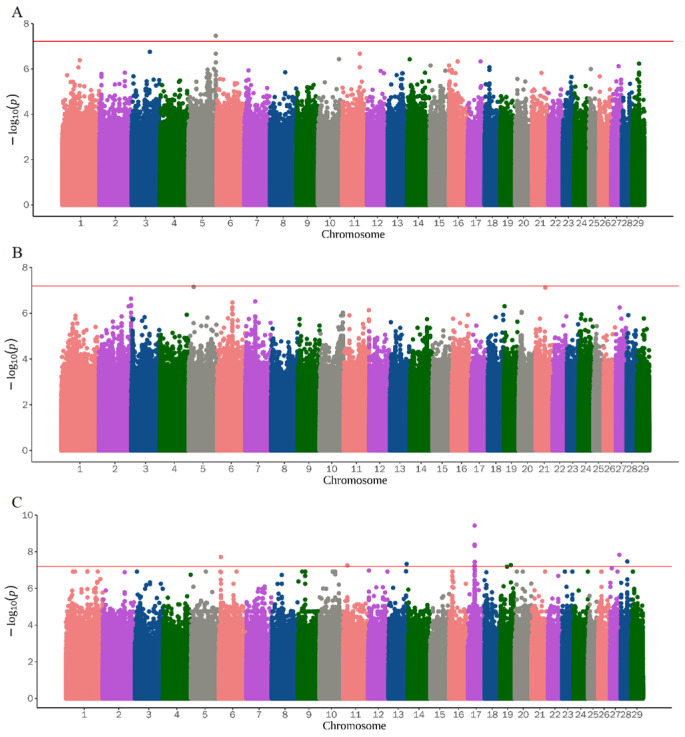
Results of GWAS for kidding traits. The red horizontal lines show the significance threshold (6.01 × 10^−8^). (**A**) ALS; (**B**) ALS3; (**C**) ALS6.

**Figure 3 genes-14-01960-f003:**
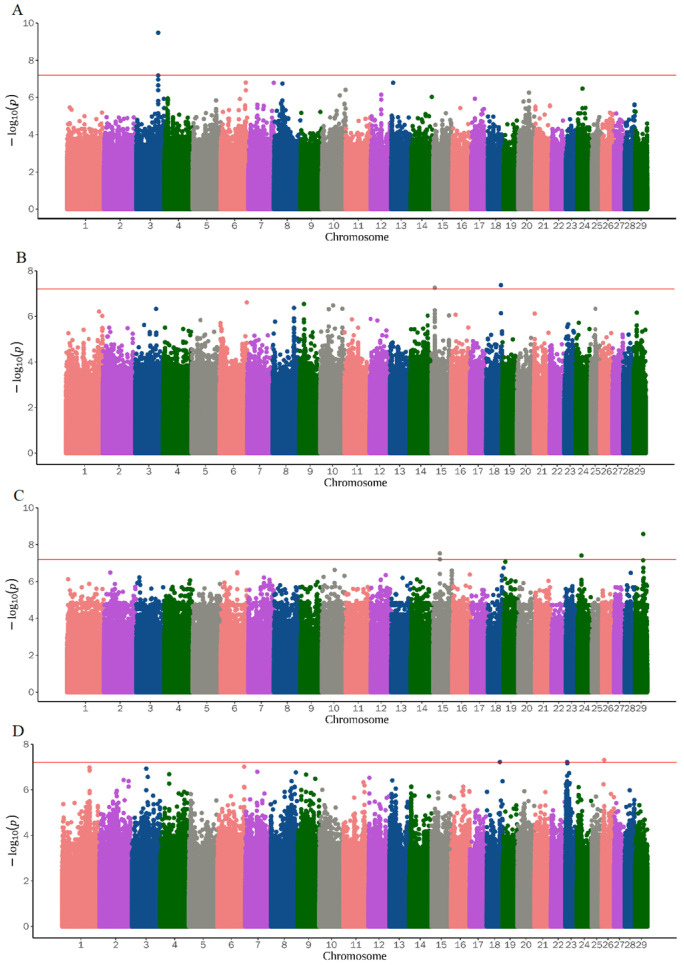
Results of GWAS for udder traits. The red horizontal lines show the significance threshold (6.01 × 10^−8^). (**A**) UD; (**B**) TD; (**C**) TL; (**D**) SNT.

**Figure 4 genes-14-01960-f004:**
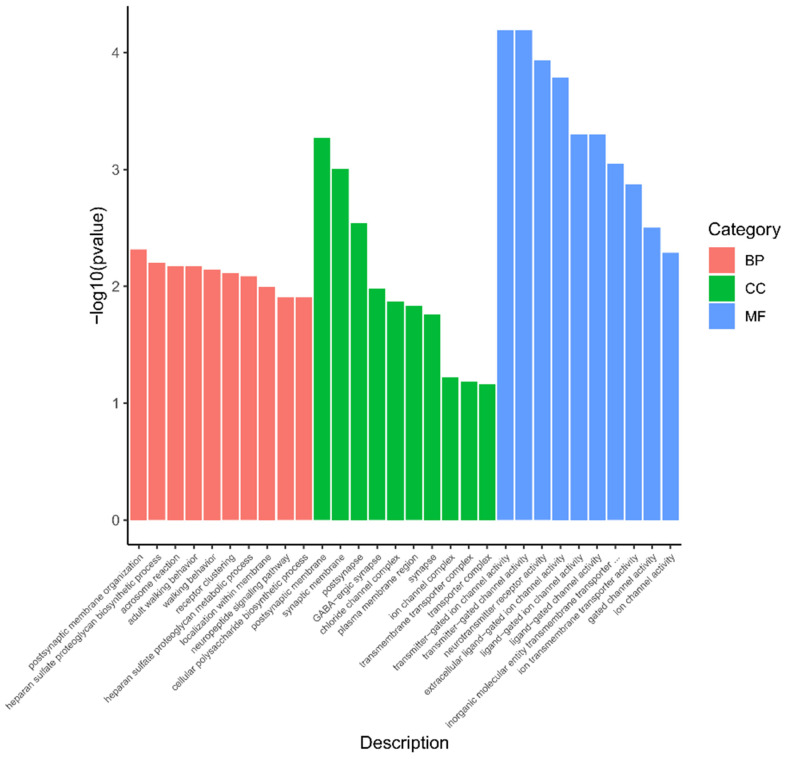
GO terms of candidate gene in average litter size in the first six parity.

**Table 1 genes-14-01960-t001:** Descriptive statistics of the reproductive traits.

Trait	Max	Min	Mean	Var	Std. Dev	CV
ALS	3	1	1.91	0.17	0.41	21.4%
ALS3	3	1	1.81	0.26	0.51	28.2%
ALS6	2.83	1.17	1.83	0.13	0.35	19.3%
UD	23.6 cm	10.6 cm	17.27 cm	8.36	2.89	16.7%
TD	26 mm	9.5 mm	15.6 mm	10.87	3.29	21.1%
TL	39.8 mm	21.1 mm	29.9 mm	22.43	4.74	15.8%
SNT	2	0	0.525	0.78	0.89	168.6%

ALS: average litter size, n = 150; ALS3: average litter size in the first three parity, n = 106; ALS6: average litter size in the first six parity, n = 50; UD: udder depth, n = 80; TD: teat diameter, n = 80; TL: teat length, n = 80; SNT: supernumerary teat, n = 80.

**Table 2 genes-14-01960-t002:** The SNPs significant associated with reproduction traits.

Trait	Chr	Position	SNP	Gene	MAF	Effect	*p*-Value
ALS	5	116147132	rs644062850	*TBC1D22A*, *TAFA5*	0.017	1.011881	3.43 × 10^−8^
ALS6	17	28987976	rs671071009	*NDST3*, *TRAM1L1*	0.090	0.7715029	3.74 × 10^−10^
	17	28985790	rs639337328	*ENSCHIG00000026285*, *ENSCHIG00000009886*	0.073	0.7564855	4.06 × 10^−9^
	17	28993318	None	*ENSCHIG00000026285*, *ENSCHIG00000009886*	0.063	0.7777119	4.74 × 10^−9^
	27	39690010	rs658833319	*ENSCHIG00000026285*, *ENSCHIG00000009886*	0.087	0.5117511	1.45 × 10^−8^
	6	7982288	rs650366329	*SLC9A8*	0.047	0.5783803	1.93 × 10^−8^
	28	28949854	rs642166030	*GLRB*	0.063	0.5344838	3.39 × 10^−8^
	17	28984146	rs655588910	*GLRB*	0.077	0.9069677	3.59 × 10^−8^
	17	29975255	rs640960816	*GLRB*	0.067	0.9069677	3.59 × 10^−8^
	17	29975258	rs656987736	*GLRB*	0.067	0.9069677	3.59 × 10^−8^
	17	29975313	rs663645959	*GLRB*	0.767	0.9069677	3.59 × 10^−8^
	13	77438831	rs640662512	*GLRB*	0.120	0.6405018	4.60 × 10^−8^
	19	45379378	rs672124332	*GRIA2*, *GASK1B*	0.157	0.4522095	5.28 × 10^−8^
	17	28986352	rs672162117	*GRIA2*, *GASK1B*	0.037	0.7584963	5.53 × 10^−8^
	11	17646846	rs651860215	*GRIA2*, *GASK1B*	0.067	0.889428	5.57 × 10^−8^
	11	17660782	rs661293290	*ENSCHIG00000027099*, *ENSCHIG00000019825*	0.080	0.889428	5.57 × 10^−8^
	11	17779280	rs661424069	*ENSCHIG00000002625*, *ENSCHIG00000020975*	0.073	0.889428	5.57 × 10^−8^
	17	29002516	None	*RHOBTB1*, *ENSCHIG00000019367*	0.093	0.4836778	6.34 × 10^−8^
UD	3	93754926	rs639924456	*ATP1A1*, *ENSCHIG00000022779*	0.493	2.180614	3.37 × 10^−10^
TD	18	62626266	None	*LRRC4C*	0.040	6.440032	4.25 × 10^−8^
	15	13770235	rs654987322	*ENSCHIG00000000547*	0.320	2.478602	5.57 × 10^−8^
TL	29	34789189	None	*SPCS2*	0.043	7.656182	2.64 × 10^−9^
	15	28785486	rs635852942	*XRRA1*	0.107	−6.364415	2.96 × 10^−8^
	24	18028701	None	*ENSCHIG00000022267*, *CELF4*	0.350	3.811532	3.90 × 10^−8^
	15	28830886	rs659007606	*NTM*, *TMEM45B*	0.123	−6.175688	6.22 × 10^−8^
SNT	26	10333285	rs649605136	*BICRA*	0.090	1.08361	4.91 × 10^−8^
	18	55634777	None	*CD83*, *ENSCHIG00000008871*	0.150	0.7668384	6.02 × 10^−8^
	23	7992077	rs662535387	*ATE1*, *FGFR2*	0.070	1.119172	6.11 × 10^−8^

Note: The identifiers associated with reproductive trait SNPs and their chromosomal (Chr) and base pair positions are shown here. MAF: minor allele frequency.

## Data Availability

The data presented in this study are available upon request from the corresponding author.

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
