# Peer review of "Genome-Wide Association Study of the Reproductive Traits of the Dazu Black Goat (Capra hircus) Using Whole-Genome Resequencing"

_genes, 2023, doi:10.3390/genes14101960_

Round 1

Reviewer 1 Report

The authors aim to disclose Genome-Wide Associations of the Reproductive Traits of the Dazu Black Goat.

The aim was the Dazu Black Goat breed, which presents very high prolificacy. My question is this: since the Dazu breed is very prolific, shouldn't the authors have considered comparing it with a less prolific breed in order to better understand which genes/SNPs are actually associated with the reproductive parameters under study?

The authors phenotyped some reproductive traits as the average litter size, which is a measure of breed prolificacy, however, no data were presented regarding fertility (age at first kidding or kidding interval), or kids’ weight at kidding and at weaning or number of kids weaned, kids’ mortality levels at kidding, number of kids that needed to be artificially reared, incidence of dystocia, traits of huge economic relevance.

Regarding udder traits, the authors didn’t refer to which phase of the reproductive cycle the measurements were made. It would also be interesting to correlate those measurements with milk yield traits, or at least with kids' average body gain till weaning, which is strongly correlated with goats’ milk yield potential and maternal behaviour.

Concerning the introduction/discussion, the authors referred to some studies in goats, sheep and cattle, but should be more thorough, as there are several relevant and recent studies published in sheep and goats concerning reproductive and productive traits that might be used to enrich the manuscript, e.g.:

Goat:

Reproductive traits:

·         Sun, X., Jiang, J., Wang, G., Zhou, P., Li, J., Chen, C., ... & Ren, H. (2023). Genome-wide association analysis of nine reproduction and morphological traits in three goat breeds from Southern China. Animal Bioscience, 36(2), 191.

·         Mahmoudi, P., Rashidi, A., Nazari-Ghadikolaei, A., Rostamzadeh, J., Razmkabir, M., & Huson, H. J. (2022). Genome-wide association study reveals novel candidate genes for litter size in Markhoz goats. Frontiers in Veterinary Science, 9, 1045589.

·         Al-Abri, M., Al Kharousi, K., Al Hamrashdi, A., & Salem, M. M. (2023). Genome wide association analysis for twinning ability in Jabal Akhdar Omani goats. Small Ruminant Research, 221, 106951.

·         Wijayanti, D., Zhang, S., Yang, Y., Bai, Y., Akhatayeva, Z., Pan, C., ... & Lan, X. (2022). Goat SMAD family member 1 (SMAD1): mRNA expression, genetic variants, and their associations with litter size. Theriogenology, 193, 11-19.

Udder conformation and milk traits:

·         Massender, E., Oliveira, H. R., Brito, L. F., Maignel, L., Jafarikia, M., Baes, C. F., ... & Schenkel, F. S. (2023). Genome-wide association study for milk production and conformation traits in Canadian Alpine and Saanen dairy goats. Journal of dairy science, 106(2), 1168-1189.

·         Jiang, A., Ankersmit-Udy, A., Turner, S. A., Scholtens, M., Littlejohn, M. D., Lopez-Villalobos, N., ... & Lehnert, K. (2022). A Capra hircus chromosome 19 locus linked to milk production influences mammary conformation. Journal of Animal Science and Biotechnology, 13(1), 4.

·         Teissier, M., Larroque, H., Brito, L. F., Rupp, R., Schenkel, F. S., & Robert-Granié, C. (2020). Genomic predictions based on haplotypes fitted as pseudo-SNP for milk production and udder type traits and SCS in French dairy goats. Journal of dairy science, 103(12), 11559-11573.

·         Martin, P., Palhière, I., Maroteau, C., Clément, V., David, I., Klopp, G. T., & Rupp, R. (2018). Genome-wide association mapping for type and mammary health traits in French dairy goats identifies a pleiotropic region on chromosome 19 in the Saanen breed. Journal of dairy science, 101(6), 5214-5226.

·         Mucha, S., Mrode, R., Coffey, M., Kizilaslan, M., Desire, S., & Conington, J. (2018). Genome-wide association study of conformation and milk yield in mixed-breed dairy goats. Journal of dairy science, 101(3), 2213-2225.

Sheep:

Reproductive traits:

·         Tsartsianidou, V., Pavlidis, A., Tosiou, E., Arsenos, G., Banos, G., & Triantafyllidis, A. (2023). Novel genomic markers and genes related to reproduction in prolific Chios dairy sheep: a genome-wide association study. animal, 17(3), 100723.

·         Gholizadeh, M., & Esmaeili-Fard, S. M. (2022). Multi-population joint genome-wide association study to detect genomic regions associated with litter size in sheep. Animal Production Research, 11(3), 15-26.

·         Abdoli, R., Mirhoseini, S. Z., Hossein-Zadeh, N. G., Zamani, P., Moradi, M. H., Ferdosi, M. H., & Gondro, C. (2019). Genome-wide association study of first lambing age and lambing interval in sheep. Small Ruminant Research, 178, 43-45.

·         Hernández-Montiel, W., Martínez-Núñez, M. A., Ramón-Ugalde, J. P., Román-Ponce, S. I., Calderón-Chagoya, R., & Zamora-Bustillos, R. (2020). Genome-wide association study reveals candidate genes for litter size traits in pelibuey sheep. Animals, 10(3), 434.

·         Martinez-Royo, A., Alabart, J. L., Sarto, P., Serrano, M., Lahoz, B., Folch, J., & Calvo, J. H. (2017). Genome-wide association studies for reproductive seasonality traits in Rasa Aragonesa sheep breed. Theriogenology, 99, 21-29.

·         SmoÅ‚ucha, G., Gurgul, A., Jasielczuk, I., KawÄ™cka, A., & Miksza-Cybulska, A. (2021). A genome-wide association study for prolificacy in three Polish sheep breeds. Journal of Applied Genetics, 62, 323-326.

Udder conformation and milk traits:

·         Casu, S., Usai, M. G., Sechi, T., Salaris, S. L., Miari, S., & Carta, A. (2018, February). A genome scan to detect QTL affecting udder morphology traits in dairy sheep. In Proceedings of the World Congress on Genetics Applied to Livestock Production (Vol. 11, p. 987).

·         Luigi-Sierra, M. G., Landi, V., Guan, D., Delgado, J. V., Castelló, A., Cabrera, B., ... & Amills, M. (2020). A genome-wide association analysis for body, udder, and leg conformation traits recorded in Murciano-Granadina goats. Journal of dairy science, 103(12), 11605-11617.

·         Makovický, P., Margetín, M., & Makovický, P. (2017). Estimation of genetic and phenotypic parameters for udder morphology traits in different dairy sheep genotypes. Acta Universitatis Agriculturae et Silviculturae Mendelianae Brunensis, 65(1).

·         Gutiérrez-Gil, B., El-Zarei, M. F., Alvarez, L., Bayón, Y., De La Fuente, L. F., San Primitivo, F., & Arranz, J. J. (2008). Quantitative trait loci underlying udder morphology traits in dairy sheep. Journal of dairy science, 91(9), 3672-3681.

·         McLaren, A., Kaseja, K., Yates, J., Mucha, S., Lambe, N. R., & Conington, J. (2018). New mastitis phenotypes suitable for genomic selection in meat sheep and their genetic relationships with udder conformation and lamb live weights. animal, 12(12), 2470-2479.

Methodologically, the manuscript is sound, as the group used the genomic data in a previously published paper.

The authors phenotyped continuous and discrete variables. Was this considered in the statistical model? It was not clear in the description of the statistical analysis.

The descriptive analysis (Table 1) shows the minimum, maximum and average values. It would be interesting to see the frequency of occurrence of the discrete variables, namely supernumerary teat.

The figures have poor quality.

Line 225-227

Therefore, teat number is an important consideration in goat breeding, playing a key role in the healthy growth of kids. In addition, supernumerary teats greatly affect the efficiency of machine milking.”

Supernumerary teats could be a good or a bad thing. When vestigial, it could be a problem. Moreover, machine milking systems for sheep and goats only have a place for two teats. Maybe you could discuss further the economic value added (or not) by the existence of supernumerary teats, and how it reflects in milk production potential.

Line 251

About FGFR2 roles, could be interesting to discuss what happens in domestic animals. See e.g.:

·         Ozawa, M.; Yang, Q.E.; Ealy, A.D. The expression of fibroblast growth factor receptors during early bovine conceptus develop-ment and pharmacological analysis of their actions on trophoblast growth in vitro. Reproduction 2013, 145, 191–201.

·         Bao, Z., Lin, J., Ye, L., Zhang, Q., Chen, J., Yang, Q., & Yu, Q. (2016). Modulation of mammary gland development and milk production by growth hormone expression in GH transgenic goats. Frontiers in physiology, 7, 278.

·         Spencer, T. E., Johnson, G. A., Burghardt, R. C., & Bazer, F. W. (2004). Progesterone and placental hormone actions on the uterus: insights from domestic animals. Biology of reproduction, 71(1), 2-10.

·         Spencer, T. E., & Bazer, F. W. (2004). Conceptus signals for establishment and maintenance of pregnancy. Reproductive Biology and Endocrinology, 2, 1-15.

Minor suggestions can be found in the attached pdf file.

Minor review are needed.

Author Response

Dear Editor and Reviewers,

On behalf of my co-authors, we are very grateful to you for allowing us to major revise our manuscript. We considered all comments and revised the manuscript entitled “Genome-Wide Association Study of the Reproductive Traits of the Dazu Black Goat (Capra hircus) using Whole Genome Resequencing”. We corrected the manuscript as follows:

Reviewer #1:

  1. The aim was the Dazu Black Goat breed, which presents very high prolificacy. My question is this: since the Dazu breed is very prolific, shouldn't the authors have considered comparing it with a less prolific breed in order to better understand which genes/SNPs are actually associated with the reproductive parameters under study?

Response:

Thank you for pointing this out. Although Dazu black goats have high fecundity, there are still individuals with low fecundity in the population, so only reproductive traits of Dazu Black Goats were investigated in this study. In subsequent studies, comparisons will be made with low fecundity goat breeds for a more accurate understanding of genes/SNPs associated with reproductive traits.

  1. The authors phenotyped some reproductive traits as the average litter size, which is a measure of breed prolificacy, however, no data were presented regarding fertility (age at first kidding or kidding interval), or kids’ weight at kidding and at weaning or number of kids weaned, kids’ mortality levels at kidding, number of kids that needed to be artificially reared, incidence of dystocia, traits of huge economic relevance.

Response:

Thank you for pointing this out. I agree with this comment. The research objective of this thesis is to excavate candidate genes for high breeding performance of Dazu Black Goat. These traits have not been collected, so these traits have not been studied in this study. Therefore, these traits could not be studied, and we will add these traits in the subsequent study.

  1. Regarding udder traits, the authors didn’t refer to which phase of the reproductive cycle the measurements were made. It would also be interesting to correlate those measurements with milk yield traits, or at least with kids' average body gain till weaning, which is strongly correlated with goats’ milk yield potential and maternal behaviour. Concerning the introduction/discussion, the authors referred to some studies in goats, sheep and cattle, but should be more thorough, as there are several relevant and recent studies published in sheep and goats concerning reproductive and productive traits that might be used to enrich the manuscript.

Response:

Thank you for pointing this out. I agree with this comment. The period of udder traits determination has been supplemented in this article (Line 82). At present, the milk yield and daily gain have not been measured, so this study has not carried out the research in this area, and this part of the character will be studied in the following research. The introduction and discussion have been enriched as required (Line 63-66, 278-284).

  1. The authors phenotyped continuous and discrete variables. Was this considered in the statistical model? It was not clear in the description of the statistical analysis.

Response:

Thank you for pointing this out. I agree with this comment. This is taken into account in statistical analysis. This study references previous research ideas and analytical methods. The statistical models in this study were referenced to those already reported in previous studies. The model is a correction for positive group structure and complex affinities within groups. And before the analysis, we carried out quality control on the phenotypic data and eliminated the abnormal phenotype (Line 103-107).

  1. The descriptive analysis (Table 1) shows the minimum, maximum and average values. It would be interesting to see the frequency of occurrence of the discrete variables, namely supernumerary teat.

Response: Thank you for pointing this out. I agree with this comment. I have added the frequency of occurrence of the supernumerary teat in the text (Line 134-136).

  1. The figures have poor quality.

Response:

Thank you for pointing this out. I agree with this comment. And the figures have been replaced.

  1. “Therefore, teat number is an important consideration in goat breeding, playing a key role in the healthy growth of kids. In addition, supernumerary teats greatly affect the efficiency of machine milking.”

Supernumerary teats could be a good or a bad thing. When vestigial, it could be a problem. Moreover, machine milking systems for sheep and goats only have a place for two teats. Maybe you could discuss further the economic value added (or not) by the existence of supernumerary teats, and how it reflects in milk production potential.

Response:

Thank you for pointing this out. I agree with this comment. And the corresponding content has been added in the text (Line 300-302)

  1. About FGFR2 roles, could be interesting to discuss what happens in domestic animals. See e.g.

Response:

Thank you for pointing this out. I agree with this comment. And corresponding content has been added in the text (Line 322-324)

  1. Minor suggestions can be found in the attached pdf file.

Response:

Thank you for pointing these out. All minor recommendations have been modified in the text

We believe we have justified the major concerns of Reviewer 1. We thank all reviewers for their time, effort, and valuable suggestions, which have enabled us to improve the manuscript further. We are looking forward to the other minor points from both reviewers if any.

With kind regards,

Xingqiang Fang

11th Oct.

Reviewer 2 Report

The paper deals with the GWAS of reproductive traits in goats. Some revisions are needed as follows.

Introduction:

Give for the breed the reproduction percentage for the flock in per cent.

Chongqing is Chinese province?

Dazu Black Goat is dairy or meat breed?

Give the heritability of litter size in goat.

Materials and Methods:

At which age were evaluated the udder traits?

Important notice: describe the principle of the genome analysis.

Results:

I did not have the Supplementary materials at disposal, why?

Table 1, you give the maximal litter size of 2.83, and the minimal of 1.17, please correct. Describe the abbreviations of statistics in legend.

Figures are completely unreadable and must be re-arranged.

Table 2, three decimals for MAF. Add legend.

Figure 4 should be 3. Again unreadable.

Discussion:

In section you focus the comparing of genes with other authors. What about the comparing of the similarity-dissimilarity of genome regions? There are some recent GWAS studies of interest:

Sun et al. DOI: 10.3390/genes14061183,

Al-Abri et al. DOI: 10.1016/j.smallrumres.2023.106951,

Easa et al. DOI: 10.3390/genes13101773, e.g.

Formal comments:

Complete the address for affiliation 1.

Abstract, r. 18, use word goat, not capra.

Author Response

Dear Editor and Reviewers,

On behalf of my co-authors, we are very grateful to you for allowing us to major revise our manuscript. We considered all comments and revised the manuscript entitled “Genome-Wide Association Study of the Reproductive Traits of the Dazu Black Goat (Capra hircus) using Whole Genome Resequencing”. We corrected the manuscript as follows:

Reviewer #2:

Introduction:

  1. Give for the breed the reproduction percentage for the flock in per cent.

Response:

Thank you for pointing this out. I agree with this comment. Therefore, I have added the reproduction percentage to the text. (Line 31-34)

  1. Chongqing is Chinese province?

Response:

Thank you for pointing this out. Chongqing is a municipality directly under the central government of China and a provincial administrative unit

  1. Dazu Black Goat is dairy or meat breed?

Response:

Thank you for pointing this out. Dazu black goat is an excellent local goat breed for both meat and skin.

  1. Give the heritability of litter size in goat.

Response:

Thank you for pointing this out. I agree with this comment. The heritability of litter size in goats is low and varies from one goat breed to another. And the content about the heritability of litter size in goats has been added to the text.

Materials and Methods:

  1. At which age were evaluated the udder traits?

Response:

Thank you for pointing this out. I agree with this comment. All the goats are 3-5 years old. And the period of udder traits determination has been supplemented in this article (Line 82).

  1. Important notice: describe the principle of the genome analysis.

Response:

Thank you for pointing this out. I agree with this comment. And the principle of the genome analysis has been added in the text (Line 97-104).

Results:

  1. I did not have the Supplementary materials at disposal, why?

Response:

Thank you for pointing this out. Supplementary material is in a separate document that I will re-upload.

  1. Table 1, you give the maximal litter size of 2.83, and the minimal of 1.17, please correct. Describe the abbreviations of statistics in legend.

Response:

Thank you for pointing this out. The maximum and minimum litter size values in the table are the average of the first six parity and therefore have decimals. Abbreviations are described in the table notes.

  1. Figures are completely unreadable and must be re-arranged.

Response:

Thank you for pointing this out. I agree with this comment. And the figures have been replaced.

  1. Table 2, three decimals for MAF. Add legend.

Response:

Thank you for pointing this out. I agree with this comment. Changes have been made to the table as required (Line 170)

  1. Figure 4 should be 3. Again unreadable.

Response:

Thank you for pointing this out. I agree with this comment. And the figures have been replaced.

Discussion:

  1. In section you focus the comparing of genes with other authors. What about the comparing of the similarity-dissimilarity of genome regions? There are some recent GWAS studies of interest:

Response:

Thank you for pointing this out. I agree with this comment. And the corresponding content has been added in the text (Line 282-285).

Formal comments:

  1. Complete the address for affiliation 1.

Response:

Thank you for pointing this out. I agree with this comment. And addresses have been replenished

  1. Abstract, r. 18, use word goat, not capra

Response:

Thank you for pointing this out. I agree with this comment. The word Capra has been deleted.

We believe we have justified the major concerns of Reviewer 2. We thank all reviewers for their time, effort, and valuable suggestions, which have enabled us to improve the manuscript further. We are looking forward to the other minor points from both reviewers if any.

With kind regards,

Xingqiang Fang

11th Oct.

Reviewer 3 Report

REVIEW:

 The manuscript titled: Genome-Wide Association Study of the Reproductive Traits of the Dazu Black Goat (Capra hircus) using Whole Genome Resequencing by X. Fanget al.

The article is describe GWAS study on the  reproductive characteristics, at the genome level. Reproductive traits belong to the important economic characteristics not only in goat breeding, but in farm animal breeding.

The litter size is affected by multiple factors and is a very important economic trait.

Phenotypic characteristics are currently monitored at the genomic level and the obtained results provide biological information for the improvement of properties (reproductive, physiological, health) in the breeding of farm animals, in this case it is the breeding of goats, and their application in the genomic selection of animals. The article has appropriate paragraphs, i.e. introduction, materials, statistics, results and discussion. The article needs to be improve or re-write for a short communication.

 QUESTIONS:

 1. Statistical analysis – needs to be improved; The author is a co-author of the publication Gu et al. 2022, GWAS of body conformation traits by whole genome sequencing in black Dazu goats - I recommend adding a link to this article - add reference.

2, enlarge pictures 1-3, as well as labels - description - is illegible

3, table 2 explain the abbreviations under the table and unite with table 1 and throughout the article

4, in the Discussion paragraph, I am missing a small summary - see in the aforementioned article - the last paragraph of the discussion94L- what data? observed or significant data?

106L - what is the reference?

120-121L - how do you explain it?

123L- do you also count genetic variation as part of the statistical analysis? if so, can you add the data to Table 2?

123L- Figure 2-3 are Manhattan diagrams and Table 2 describe calculated values and briefly comment on the results obtained

156L - what was the p-value of 0.05
